# Trends in antimicrobial resistance amongst *Salmonella* Paratyphi A isolates in Bangladesh: 1999–2021

**Mohammad S. I. Sajib**[1], **Arif M. Tanmoy**[1], **Yogesh Hooda**[1], **Hafizur Rahman**[1], **Sira J. Munira**[1], **Anik Sarkar**[1], **Dipu Das**[1], **Md. Asadur Rahman**[2], **Nazrul Islam**[1], **Mohammod Shahidullah**[3], **Md. Ruhul Amin**[4], **Md. Jahangir Alam**[4], **Mohammed Hanif**[4], **Stephen P. Luby**[5], **Denise O. Garrett**[6], **Samir K. Saha**[1,7] *, **Senjuti Saha**[1] *

**1** Child Health Research Foundation, Dhaka Bangladesh, **2** Popular Diagnostic Center, Dhaka, Bangladesh, **3** Department of Neonatology, Bangabandhu Sheikh Mujib Medical University, Dhaka, Bangladesh, **4** Department of Pediatrics, Bangladesh Institute of Child Health, Dhaka, Bangladesh, **5** Division of Infectious Diseases and Geographic Medicine, Stanford University School of Medicine, Stanford, California, United States of America, **6** Sabin Vaccine Institute, Washington, DC, United States of America, **7** Department of Microbiology, Bangladesh Shishu Hospital and Institute, Dhaka, Bangladesh

* samir@chrfbd.org (SKS); senjutisaha@chrfbd.org (SS)

**Data Availability Statement:** All data is present in the manuscript and its supporting files.

## Abstract

### Background

Typhoid and paratyphoid remain common bloodstream infections in areas with suboptimal water and sanitation infrastructure. Paratyphoid, caused by *Salmonella* Paratyphi A, is less prevalent than typhoid and its antimicrobial resistance (AMR) trends are less documented. Empirical treatment for paratyphoid is commonly based on the knowledge of susceptibility of *Salmonella* Typhi, which causes typhoid. Hence, with rising drug resistance in *Salmonella* Typhi, last-line antibiotics like ceftriaxone and azithromycin are prescribed for both typhoid and paratyphoid. However, unlike for typhoid, there is no vaccine to prevent paratyphoid. Here, we report 23-year AMR trends of *Salmonella* Paratyphi A in Bangladesh.

### Methods

From 1999 to 2021, we conducted enteric fever surveillance in two major pediatric hospitals and three clinics in Dhaka, Bangladesh. Blood cultures were performed at the discretion of the treating physicians; cases were confirmed by culture, serological and biochemical tests. Antimicrobial susceptibility was determined following CLSI guidelines.

### Results

Over 23 years, we identified 2,725 blood culture-confirmed paratyphoid cases. Over 97% of the isolates were susceptible to ampicillin, chloramphenicol, and cotrimoxazole, and no isolate was resistant to all three. No resistance to ceftriaxone was recorded, and >99% of the isolates were sensitive to azithromycin. A slight increase in minimum inhibitory concentration (MIC) is noticed for ceftriaxone but the current average MIC is 32-fold lower than the

**Funding:** This work was funded by the Gavi Pneumococcal Vaccines Accelerated Development and Introduction Plan (PneumoADIP, grant number not available) to SKS, the World Health Organization, Switzerland, Invasive Bacterial Vaccine Preventable Diseases study (grant numbers 201588766, 201233523, 201022732, 200749550) to SKS, and Bill and Melinda Gates Foundation, USA, SEAP study (grant number INV-008335) to SKS, SS and DOG. The funders had no role in study design, data collection and analysis, decision to publish, or preparation of the manuscript.

**Competing interests:** The authors declare no competing interests.

resistance cut-off. Over 99% of the isolates exhibited decreased susceptibility to ciprofloxacin.

## Conclusions

*Salmonella* Paratyphi A has remained susceptible to most antibiotics, unlike *Salmonella* Typhi, despite widespread usage of many antibiotics in Bangladesh. The data can guide evidence-based policy decisions for empirical treatment of paratyphoid fever, especially in the post typhoid vaccine era, and with the availability of new paratyphoid diagnostics.

## Author summary

Typhoid and paratyphoid fever, caused by *Salmonella* Typhi and Paratyphi A respectively, are common in areas lacking safe water and optimum infrastructure. With increasing multidrug resistance in *Salmonella* Typhi, newer antimicrobials like ceftriaxone and azithromycin are frequently prescribed for the empirical treatment of both typhoid and paratyphoid fever. This is because *Salmonella* Paratyphi A is less prevalent in most endemic countries, and typhoid and paratyphoid fever are often symptomatically indistinguishable. In this study, we conducted comprehensive surveillance of *Salmonella* Paratyphi A in Bangladesh over a period of 23 years and identified 2,725 blood culture-confirmed paratyphoid cases. Our findings indicate that *Salmonella* Paratyphi A remained susceptible to most of the older generation of antibiotics as over 97% of the isolates continue to be susceptible to ampicillin, chloramphenicol and cotrimoxazole, and no isolate detected was resistant to all three drugs. Although 99% of the isolates exhibited reduced fluoroquinolone susceptibility, 99% and 100% were sensitive to azithromycin and ceftriaxone. These findings are important and can guide evidence-based policy decisions for the empirical treatment of paratyphoid fever. While newer antimicrobials such as azithromycin and ceftriaxone remain effective, successful implementation of typhoid conjugate vaccines and rapid diagnostics may re-enable the use of older generation antibiotics for empirical treatment of enteric fever without compromising the treatment efficacy. This approach can contribute to the antimicrobial stewardship efforts by preserving critically important antimicrobials such as ceftriaxone and azithromycin for the treatment of numerous other multidrug-resistant bacterial infections.

## Introduction

Typhoid and paratyphoid fever, collectively known as enteric fever is estimated to have caused 14 million illnesses and 136 thousand deaths in 2017 [1]. The burden is disproportionately high in low- and middle-income countries in Asia and Africa [1–3]. In 2018, WHO prequalified the first typhoid conjugate vaccine (TCV), Vi-tetanus toxoid (Vi-TT), for routine immunization of children over 6 months. This single-dose conjugate vaccine provides immunity against *Salmonella* Typhi, the causative agent of typhoid fever, and has been recommended by the Strategic Advisory Group of Experts for routine use in countries where typhoid fever is endemic [4,5]. Studies from Nepal, Bangladesh, India, Pakistan and Malawi have shown high efficacy of Vi-TT vaccine in preventing typhoid fever [6–10]. In 2020 a second TCV, Vi-diphtheria toxin (Vi-CRM$_{197}$) vaccine, was prequalified by WHO and clinical trials are ongoing [11,12]. However, both TCVs are specific to *Salmonella* Typhi and provide no cross-protection

against *Salmonella* Paratyphi A, a related pathogen that causes the symptomatically similar but less prevalent disease, paratyphoid fever. About 20% of enteric fever cases are that of paratyphoid [1,2,13,14]. No vaccines are available yet to protect against paratyphoid fever although there are a few candidates, live attenuated (CVD1902) or conjugated (O:2-TT and $CRM_{197}$), that are currently in the pipeline [15]. Therefore, as the number of typhoid fever cases decline substantially in the post-TCV period, the proportions of typhoid and paratyphoid fever cases are expected to change, like seen in certain geographical locations [16].

Paratyphoid fever can lead to severe morbidity, and even death if treatment is delayed or inappropriate [2]. It is a major cause of febrile illness in South and Southeast Asia, and sub-Saharan Africa, estimated to cause more than 3.3 million cases a year with 19 thousand deaths globally [1]. To date, *Salmonella* Typhi has been substantially more common and exerted a greater burden of disease than *Salmonella* Paratyphi A [13, 17]. In Bangladesh, a country with a very high burden of enteric fever, one out of six enteric fever cases is caused by *Salmonella* Paratyphi A and this proportion of paratyphoid cases has been consistent over the last two decades [13]. The relatively low number of cases of paratyphoid fever compared to typhoid fever has contributed to neglect over the years, resulting in a paucity of data on epidemiology and antimicrobial resistance (AMR) of paratyphoid fever. The empirical treatment for paratyphoid is commonly based on the available antimicrobial susceptibility data of the more prevalent *Salmonella* Typhi [18,19].

In contrast to the plethora of reports on antimicrobial resistance (AMR) in *Salmonella* Typhi, there are only a handful of studies focused exclusively on *Salmonella* Paratyphi A [20,21], and most of them are limited by short study duration, a small number of cases, and a lack of multi-modality surveillance platform [13,22]. In the post-TCV-era, if typhoid fever cases begin to fall, it will be important to adjust empirical treatment regimens based on the data on AMR patterns of *Salmonella* Paratyphi A. In this study, our primary objective was to characterize the prevalence and trends of AMR in *Salmonella* Paratyphi A to commonly used antimicrobials, using quantitative data from disc diffusion and broth microdilution. We analyzed the data of 2,725 *Salmonella* Paratyphi A isolates collected from multiple sites in Dhaka from 1999 through 2021, making this one of the largest studies on trends and patterns of AMR of *Salmonella* Paratyphi A.

## Methods

### Ethics statement

The protocols were approved by the ethics review committees of the Bangladesh Shishu Hospital and Institute. For the hospitalized cases, informed written consent and clinical information were taken from parents/legal guardians of participants. For out-patient cases no formal consent was obtained as blood samples were collected as part of routine clinical care at the discretion of the treating physician and data on *Salmonella* Paratyphi A positive cases were retrospectively included in this study from the clinical records with no identifiable information.

### Surveillance settings

The enteric fever surveillance was initiated in Dhaka, Bangladesh in 1999. Considering the different epidemiological characteristics, complex care-seeking and treatment behavior in Bangladesh described earlier [22], we included three types of health care facilities across five sites in Dhaka, where enteric fever burden is estimated to be the highest [14]: (a) Bangladesh Shishu Hospital and Institute (BSHI, previously known as Dhaka Shishu Hospital or DSH), (b) Dr. M R Khan Shishu Hospital & Institute of Child Health (previously known as Shishu Shasthya

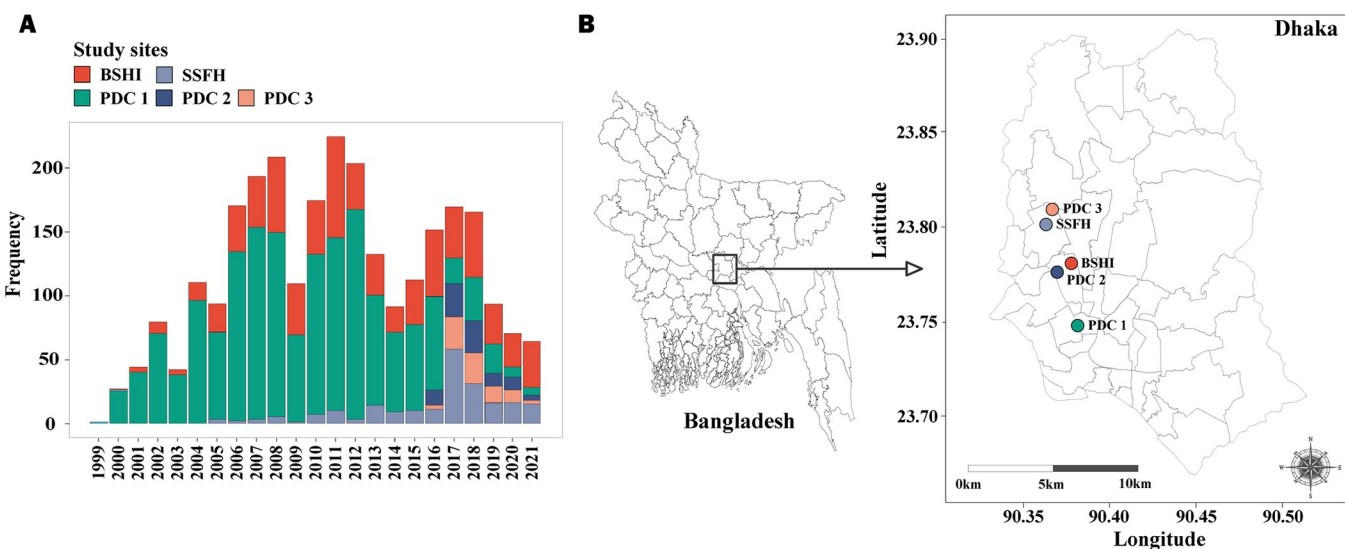

**Fig 1. Yearly Distribution of Salmonella Partyphi A Detected at the 5 Surveillance Sites in Dhaka, Bangladesh. (A)** Number of *Salmonella* Paratyphi A isolates collected between 1999 and 2021 from the study sites. **(B)** A map of Dhaka, Bangladesh showing all the study sites. The map was obtained from the database of Global Administrative Areas (GADM), a free and open-source database (license: https://gadm.org/license.html) through the R packages maps and maptools.

Foundation Hospital or SSFH), and (c) three branches of an outpatient-based clinic Popular Diagnostic Center (PDC1, PDC2, and PDC3) (Fig 1B). PDC2 and PDC3 were added as surveillance sites in October 2016. BSHI is the largest pediatric hospital in the country with 681 beds and provides primary to tertiary care to children from all over the country. This government-aided private hospital treats 38% of patients free of cost. SSFH provides primary care and is the second-largest pediatric hospital in Bangladesh with 250 beds for children, 6.5% of which are dedicated to those unable to pay for care. Popular Diagnostic Center is one of the largest private consultation centers in Bangladesh and caters to a higher socioeconomic class than the hospitals and patients of all age groups. BSHI and SSF are part of the Global Invasive Vaccine-Preventable Bacterial Disease Surveillance Network [23] and all sites have been described in detail earlier [22]. All blood samples were collected as a part of the diagnostic service and only at the attending physician's discretion.

## Isolation, identification, and antimicrobial susceptibility testing

Blood specimens were routinely subjected to standard bacteriological culture [24]. After initial isolation, identification was conducted via standard biochemical tests utilizing Klingler's Iron agar, Simmons citrate agar, motility-indole, and urea agar tests followed by agglutination with *Salmonella* Paratyphi A specific antisera (Thermo Scientific, MA, USA). Available isolates were preserved at -80°C freezer.

Results of antimicrobial susceptibility testing using the Kirby Bauer disk diffusion method for ampicillin, chloramphenicol, cotrimoxazole, ciprofloxacin, ceftriaxone, and azithromycin were extracted from available electronic records at the Child Health Research Foundation (CHRF), who operate the microbiological diagnostic laboratories in the two hospitals, and store data digitally. Antibiotic susceptibility data and isolates were also obtained from PDC branches. While electronic records of all the samples tested or reported as *Salmonella* Paratyphi A were readily accessible at the CHRF, historical data for PDC branches up until 2016 mostly existed in paper format and were available only for culture positive samples. Some of

these reports, from CHRF (BSHI and SSFH) and mainly from PDC1, 2 and 3 had data missing on AST or other variables. However, as majority of these historical isolates were already preserved in our biobank, it was decided to revive them and repeat AST for the missing antimicrobials. During the re-testing, isolates that were resistant to azithromycin by the disc diffusion method, were retested using MIC strips [bioMérieux, Marcy-l'Étoile, France]. Finally, all the resuscitated isolates, historical or contemporary, were tested to determine the minimum inhibitory concentrations (MICs) of ciprofloxacin and ceftriaxone using the broth microdilution method [25]. MIC determination provides a greater resolution to observe subtle changes in susceptibility amongst non-resistant isolates. Zone diameter results and MIC were interpreted according to the Clinical Laboratory Standard Institute (CLSI) guidelines [26].

### Statistical analysis

R 4.2.3 base functions, ggplot2, dplyr, maps and maptools were used for data analysis and generating figures and maps. To understand possible non-linear trend of ciprofloxacin and ceftriaxone, generalized additive model (mgcv::gam) was utilized with the formula $y \sim s(x, bs =$ "cs") for >1,000 observations per antibiotic (ciprofloxacin n = 1,205 and ceftriaxone n = 1,208). The Bangladesh district-level administrative map was obtained from the database of Global Administrative Areas (GADM), a free and open-source database (license: https://gadm.org/license.html) through the R packages maps and maptools.

## Results

Between 1999 and 2021, a total of 2,725 *Salmonella* Paratyphi A were isolated across the five different sites in Dhaka, Bangladesh (**Fig 1A**). Among them, 33.9% (n = 926) were from the hospitals and the remaining 66.1% from out-patient based consultation centers (n = 1,799) (**Fig 1A and 1B**).

For the three first line of drugs, disc diffusion data were available and/or could be generated for ampicillin (2,705, 99.2%), chloramphenicol (2,687, 98.6%), and cotrimoxazole (2,713, 99.5%) isolates. Overall, 97.4% (2,635/2,705) of the strains were susceptible to ampicillin, 99.7% (2,680/2,687) to chloramphenicol and 99% (2,687/2,713) to cotrimoxazole. No single isolate was resistant to all three first-line drugs at the same time, indicating no circulation of multi-drug resistant (MDR) *Salmonella* Paratyphi A isolates in Dhaka, Bangladesh. Disc diffusion data for ceftriaxone could be gathered and generated for 2,391 (87.7%) of the isolates, for 1,556 (57.1%) of the isolates for ciprofloxacin. No resistance to ceftriaxone was detected. However, 98.9% (1,540/1,556) of the isolates were non-susceptible to ciprofloxacin (**Fig 2**).

To probe further into the trend of resistance or susceptibility of ciprofloxacin and ceftriaxone, we attempted to revive the bio-banked isolates and conduct micro broth dilution tests to measure the MIC of these two antibiotics. As all isolates were non-resistant to ceftriaxone, and most to ciprofloxacin, MIC determination would provide greater resolution to observe subtle changes in susceptibility amongst non-resistant isolates. Decreased ciprofloxacin susceptibility was noted in 1,192 of 1,205 (98.9%) that could be revived and tested. MIC50 (concentration that inhibits ≥50% of the isolates) was 0.5 µg/mL and MIC90 (concentration that inhibits ≥90% of the isolates) was also 0.5 µg/mL. 90.6% (1,092/1,205) isolates were found to be intermediate resistant (MIC <1 µg/mL) and 8.2% (100/1,205) were found to be completely resistant (MIC > 1 µg/mL). Average MIC has remained consistent since 2005, ranging between 0.25 to 0.5 µg/mL till 2021 **Fig 3A**). We also tested 1,208 isolates to obtain the MIC (MIC50 = 0.0625 µg/mL, MIC90 = 0.125 µg/mL) of ceftriaxone; all MIC values were much lower than the cut-off for resistance of 4 µg/mL (**Fig 3B**). A slight increase in MIC is noticed over the years but the average MIC for the last 2 years of the study, 2020 and 2021, is 0.125 µg/

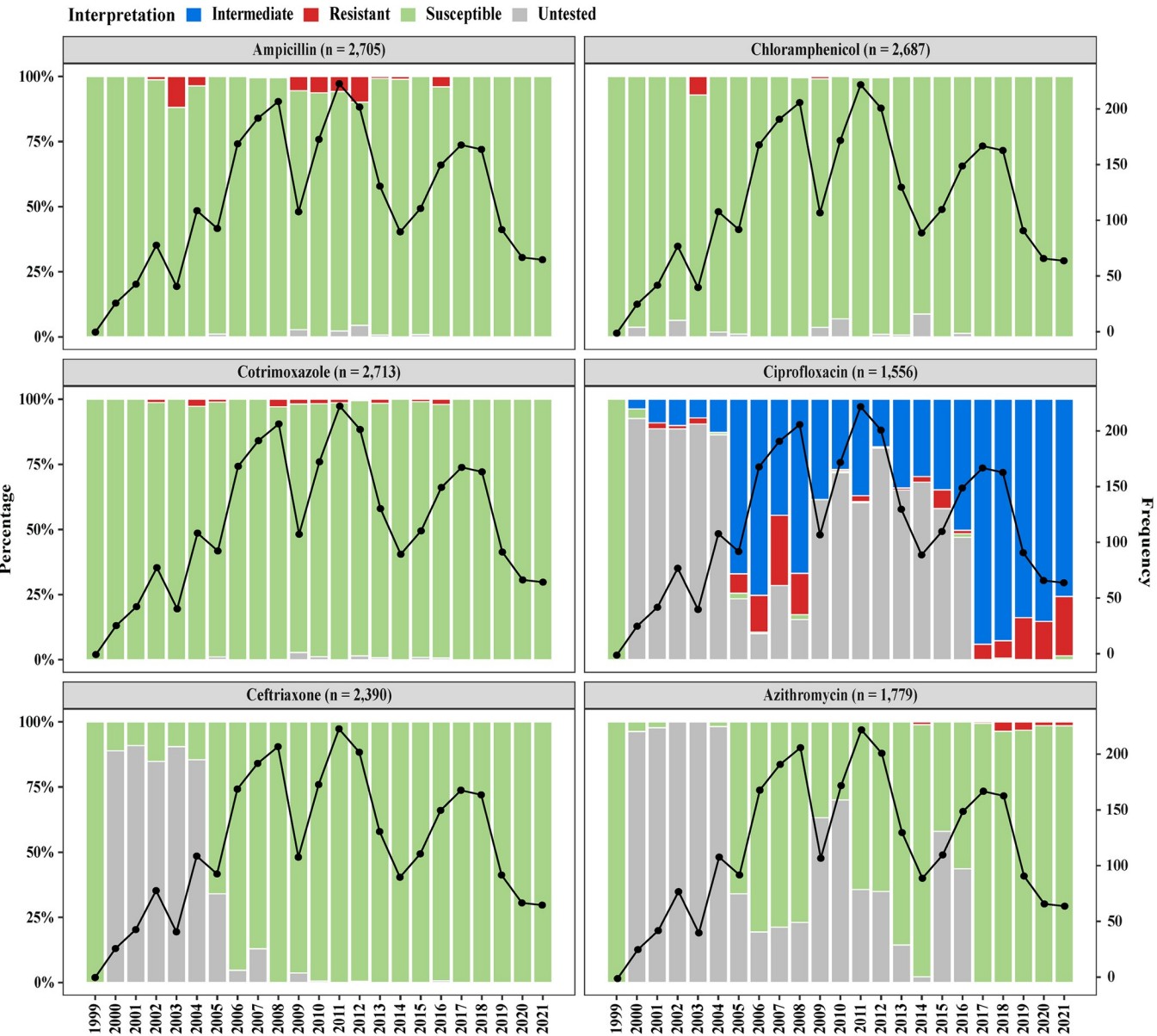

**Fig 2. Antimicrobial susceptibility pattern of *Salmonella* Paratyphi A against ampicillin (n = 2,705), chloramphenicol (n = 2,687), cotrimoxazole (n = 2,713), ciprofloxacin (n = 1,556), ceftriaxone (n = 2,390), and azithromycin (n = 1,779) using disc diffusion.** The primary y axis depicts the proportions of intermediate (blue), resistant (red), susceptible (green) and untested isolates (grey; isolates that could not be tested/revived and data for some antimicrobials were not available in the record). The black dotted line in the secondary y axis is showing the frequency of *Salmonella* Paratyphi A cases observed each year during the study period (total n = 2,725).

mL, which is 32 folds lower than the resistance cut-off of according to 4 μg/mL according to CLSI (**Fig 3B**).

The first confirmed case of azithromycin-resistant *Salmonella* Paratyphi A was detected in 2014 from the same surveillance network [27]. Disc diffusion data for azithromycin were available for 1,779 isolates. However, previous work has shown that disc diffusion method can detect azithromycin susceptible strains with 100% specificity and 100% sensitivity, but the sensitivity of detection of resistant strains is much lower [28,29]. Therefore, all isolates interpreted as resistant using disc diffusion were retested using MIC strips. In total, of 1,779 isolates tested,

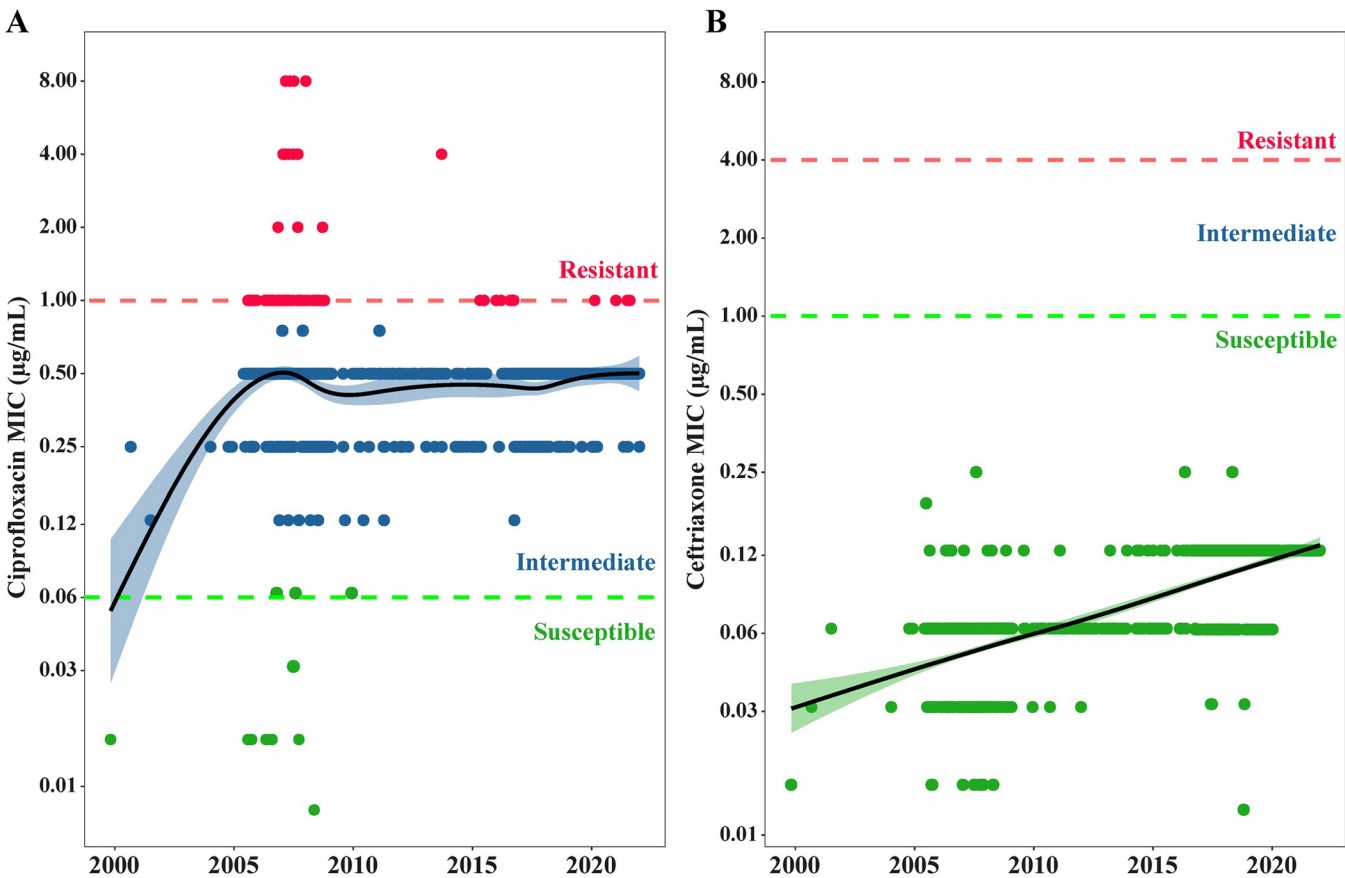

**Fig 3.** Minimum inhibitory concentration of *Salmonella* Paratyphi A to **(A)** ciprofloxacin (n = 1,205) and **(B)** ceftriaxone (n = 1,208) isolated from 1999 to 2021 in Bangladesh. The colored dots [red (resistant), blue (intermediate), and green (susceptible)] represent the MIC value of an isolate during a specific period. Generalized additive model (black line) with a 95% confidence interval (blue and green shades) is describing the changes in MICs against these antimicrobials over time. The horizontal dotted lines (red; resistant, green: susceptible) are the antibiotic cut-off limits according to CLSI-2022. Each individual dots corresponds to a single isolate and its color highlights the interpretation of phenotypic resistance as per CLSI-2022 guidelines.

1,766 (99.3%) were susceptible to azithromycin; only 13 strains (0.7%) were resistant (mean MIC 151.38 μg/mL).

## Discussion

This study reports one of the largest series of *Salmonella* Paratyphi A (n = 2,725) isolates collected through multiple surveillance modalities that includes both outpatient and inpatient departments, during a 23-year period between 1999 and 2021. Most studies on paratyphoid fever have been sporadic in South Asia, and the burden of disease and associated AMR depicted by these studies has been variable [3,30]. A reason for this may be that in South Asia most of the enteric fever cases are treated at home, and cases are only admitted in the hospital in case of treatment failure and complications, which leads differences in burden of disease and AMR patterns depending on whether the studies were conducted in hospitals or communities [13,18]. We have previously reported on the importance of including different health facilities to ensure comprehensive data capture [22]. In this study, we have included paratyphoid cases from both community clinics and hospitals to ensure comprehensive coverage (S1 and S2 Tables).

No isolate of *Salmonella* Paratyphi A exhibited multi-drug resistance, i.e., resistance to all three first-line drugs, amoxicillin, cotrimoxazole, and chloramphenicol. Individual resistance

to any of these drugs individually was also very rare, and 96.7% of isolates were sensitive to all three. These findings are concordant with other smaller reports from South Asia [31–33]. Resistance to these drugs is usually carried in plasmids, and a recent large-scale genomic analysis of *Salmonella* Paratyphi A showed that the bacterium does not commonly carry such plasmids [34]. In contrast, more than 98% of the *Salmonella* Paratyphi A isolates were non-susceptible to fluoroquinolone. This was likely driven by overtreatment and irrational use of quinolones to treat enteric fever all over South Asia and is often mediated by single point mutations in quinolone-resistance determining region (QRDR) [34–38].

Most endemic countries lack the facility or the resources to conduct blood culture and antimicrobial susceptibility testing [24,39]. The majority ($>$ 90%) of enteric fever patients are treated in the community empirically, and previous work has shown that for each confirmed case of typhoid fever, 3–25 patients without a confirmed detection of typhoidal *Salmonella* are treated with antimicrobials [18]. Prescriptions commonly include a newer generation of drugs like ceftriaxone and azithromycin. In our study, no resistance was detected against third-generation cephalosporin and ceftriaxone. To date, there is only one published report of a ceftriaxone resistant Paratyphi A isolate and that was detected in the UK from a Bangladeshi traveler [40]. Other studies also found no ceftriaxone resistance in *Salmonella* Paratyphi A [30,33,35,37]. Examination of minimum inhibitory concentration depicted a slightly increasing trend, but the average MIC is still 32 folds lower than the MIC required to render it completely resistant to ceftriaxone. As of 2021, <1% of the isolates exhibited resistance to azithromycin in Bangladesh. The first azithromycin resistant isolate was detected in 2014; resistance to azithromycin is caused by spontaneous point mutations in the *acrB* gene [27,41]. This is concordant with reports from other countries in the region [42–46].

Taken together these findings suggest that empirical treatment of paratyphoid fever with ceftriaxone and azithromycin will continue to be effective. However, our findings also suggest that a successful implementation of TCV could facilitate a shift in the approach to enteric fever treatment. If *Salmonella* Typhi becomes less prevalent during post-TCV era, and if there is increased availability of rapid diagnostics to detect paratyphoid fever [47] there is potential to base empirical treatment on the susceptibility patterns of *Salmonella* Paratyphi A. Since none of the *Salmonella* Paratyphi A in Dhaka exhibited multidrug resistance, first line of drugs like ampicillin, cotrimoxazole, or chloramphenicol can be used empirically for paratyphoid fever without affecting the treatment efficacy. This would allow us to reserve ceftriaxone and azithromycin for the treatment of other severe bacterial infections and greatly benefit antimicrobial stewardship efforts in Bangladesh.

The findings of the analysis presented here should be considered within the context of a few limitations. The cases presented here represent only a small proportion of the people with enteric fever who come into health facilities for blood culture. In addition, blood culture sensitivity is 61% in detected cases, indicating, many cases were missed [48]. This is also illustrated by the detection of the ceftriaxone resistant case in UK from a traveler to Bangladesh [38] that even the large surveillance system presented here is not comprehensive. Finally, the unavailability of data of total blood cultures done at all sites prohibit analysis of trends on overall blood culture positivity of *Salmonella* Paratyphi A.

In summary, this study presents a systematic surveillance of paratyphoid fever in Bangladesh over 23 years in multiple health care facilities and highlights that barring fluoroquinolones, *Salmonella* Paratyphi A has remained sensitive to most classes of antibiotics including the first line. However, it is imperative to remain vigilant and monitor patterns of resistance through surveillance systems. Evidence-based rational use of antibiotics to treat paratyphoid fever, in addition to continued improvements of water sanitation and hygiene could interrupt increase in AMR while simultaneously reducing the burden of paratyphoid in endemic countries.

## Supporting information

**S1 Table. Frequency of *Salmonella* Paratyhi A and their corresponding proportions stratified by study sites.**
(DOCX)

**S2 Table. Cases of *Salmonella* Paratyphi A and their proportional distribution across different age groups.**
(DOCX)

**S1 Data. This dataset contains the clinical and epidemiological data of all paratyphoid cases collected in our surveillance.** The following information is provided: Randomized_ID: ID of the isolates included in this study. Sites_F: Hospital/health care facility from which the sample was collected (BSHI: Bangladesh Shishu Hospital and Insitute, SSFH: Shishu Shashtya Foundation Hospital, PDC1,2 and 3: Popular Diagnostic centers). IPDOPD: Department where the sample was collected from (IPD: In-patient department; OPD: out-patient department). BloodReceiveDate: Date of blood collection. Year: Year of blood collection. Sex: Sex of the child (Male, Female, Data not found). AgeInMonths: Age of the child in months. AmpicillinZCat: Categorical reading from the disc diffusion test for Ampicillin based on CLSI guidelines (Susceptible, Intermediate, Resistant, Data not found). ChloramphenicolZCat: Categorical reading from the disc diffusion test for Chloramphenicol based on CLSI guidelines (Susceptible, Intermediate, Resistant, Data not found). CotrimoxazoleZCat: Categorical reading from the disc diffusion test for Cotrimoxazole based on CLSI guidelines (Susceptible, Intermediate, Resistant, Data not found). CiprofloxacinZCat: Categorical reading from the disc diffusion test for Ciprofloxacin based on CLSI guidelines (Susceptible, Intermediate, Resistant, Data not found). CeftriaxoneZCat: Categorical reading from the disc diffusion test for Ceftriaxone based on CLSI guidelines (Susceptible, Intermediate, Resistant, Data not found). AzithromycinZCat: Categorical reading from the disc diffusion test for Azithromycin based on CLSI guidelines (Susceptible, Intermediate, Resistant, Data not found). Isolate_available: If the Isolate was Biobanked (Yes, No). CIPMIC: MIC determined using dilution testing for Ciprofloxacin. CIPMIC_Cat: Categorical reading of the MIC test for Ciprofloxacin based on CLSI guidelines (Susceptible, Intermediate and Resistant). CROMIC: MIC determined using dilution testing for Ceftriaxone. CROMIC_Cat: Categorical reading of the MIC test for Ceftriaxone based on CLSI guidelines (Susceptible, Intermediate and Resistant).
(XLSX)

## Acknowledgments

We would like to thank Maksuda Islam, Shampa Saha, Mohammad Jamal Uddin, and all past and present members of the clinical microbiology and epidemiology team of the Child Health Research Foundation for their assistance with patient enrollment, data collection, and project management. We are also grateful to the entire SEAP team for their enthusiastic support and coordination during this project.

## Author Contributions

**Conceptualization:** Samir K. Saha.

**Data curation:** Mohammad S. I. Sajib, Hafizur Rahman, Sira J. Munira, Anik Sarkar, Dipu Das, Md. Asadur Rahman, Nazrul Islam, Senjuti Saha.

**Formal analysis:** Mohammad S. I. Sajib, Yogesh Hooda, Senjuti Saha.

**Funding acquisition:** Stephen P. Luby, Denise O. Garrett, Samir K. Saha, Senjuti Saha.

**Investigation:** Mohammad S. I. Sajib, Arif M. Tanmoy, Hafizur Rahman, Md. Asadur Rahman, Mohammod Shahidullah, Md. Ruhul Amin, Md. Jahangir Alam, Mohammed Hanif.

**Methodology:** Mohammad S. I. Sajib, Arif M. Tanmoy, Hafizur Rahman, Anik Sarkar, Dipu Das, Nazrul Islam, Samir K. Saha, Senjuti Saha.

**Project administration:** Hafizur Rahman, Sira J. Munira, Senjuti Saha.

**Supervision:** Samir K. Saha, Senjuti Saha.

**Visualization:** Mohammad S. I. Sajib, Yogesh Hooda.

**Writing – original draft:** Mohammad S. I. Sajib, Yogesh Hooda, Samir K. Saha, Senjuti Saha.

**Writing – review & editing:** Mohammad S. I. Sajib, Arif M. Tanmoy, Yogesh Hooda, Stephen P. Luby, Denise O. Garrett, Samir K. Saha, Senjuti Saha.

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
