## [Decision Letter · Decision Letter 0]

11 Aug 2023

Dear Dr. Saha,

Thank you very much for submitting your manuscript "Trends in antimicrobial resistance amongst Salmonella Paratyphi A isolates in Bangladesh: 1999-2021" for consideration at PLOS Neglected Tropical Diseases. As with all papers reviewed by the journal, your manuscript was reviewed by members of the editorial board and by several independent reviewers. The reviewers appreciated the attention to an important topic. Based on the reviews, we are likely to accept this manuscript for publication, providing that you modify the manuscript according to the review recommendations. 

Sincerely,

David Joseph Diemert, M.D.

Academic Editor

Stuart Blacksell

Section Editor

Reviewer's Responses to Questions

**Key Review Criteria Required for Acceptance?**

**Methods**

-Are the objectives of the study clearly articulated with a clear testable hypothesis stated?

-Is the study design appropriate to address the stated objectives?

-Is the population clearly described and appropriate for the hypothesis being tested?

-Is the sample size sufficient to ensure adequate power to address the hypothesis being tested?

-Were correct statistical analysis used to support conclusions?

-Are there concerns about ethical or regulatory requirements being met?

Reviewer #1: The authors set out to investigate the antimicrobial susceptibility of 2725 Salmonella Paratyphoid A isolates collected from multiple sites in Dhaka from 1999 through 2021, making this one of the largest studies on trends and patterns of AMR of Salmonella Paratyphi. The rationale is that the Paratyphoid treatment is commonly based on the available antimicrobial susceptibility data of the more prevalent Salmonella Typhi, which has extensive MDR phenotype. Lack of adequate laboratory infrastructure in endemic countries makes it difficult to sustain routine surveillance for AMR for this organism, so credible AMR data would go a long way to define reliable empiric treatment options. 

For some isolates, the team performed both Disk Diffusion and MIC determination for antimicrobial susceptibility, for others, they abstracted historical data from the archives. It is important to explain why for some antimicrobials MICs were performed while for others only disk susceptibility data was available.

Reviewer #2: -Are the objectives of the study clearly articulated with a clear testable hypothesis stated?

Yes

-Is the study design appropriate to address the stated objectives?

Yes

-Is the population clearly described and appropriate for the hypothesis being tested?

Yes 

-Is the sample size sufficient to ensure adequate power to address the hypothesis being tested?

Yes

-Were correct statistical analysis used to support conclusions?

Yes, but further methods details are required (as detailed below).

-Are there concerns about ethical or regulatory requirements being met?

No

Minor comments:

- Line 124: PDC3 is mentioned twice but I think one of those mentions should refer to different clinic.

- Line 138: Please provide a citation for the biochemical tests or a brief description.

- Line 144: I am not sure if ‘abstracted’ should be ‘extracted’.

- Figure 3: Please also add text to the methods section describing the regression techniques used to generate the black lines.

**Results**

-Does the analysis presented match the analysis plan?

-Are the results clearly and completely presented?

-Are the figures (Tables, Images) of sufficient quality for clarity?

Reviewer #1: No single isolate was resistant to all three first-line drugs at the same time, indicating no circulation of multi-drug resistant (MDR) Salmonella Paratyphi A isolates, but more than 98% of the Salmonella Paratyphi A isolates were non-susceptible to fluoroquinolone. This was likely driven by overtreatment and irrational use of quinolones to treat enteric fever all over South Asia and is often mediated by single point

mutations in quinolone-resistance determining region (QRDR). Good if the team could provide some evidence for this assertion.

Reviewer #2: -Does the analysis presented match the analysis plan?

Yes

-Are the results clearly and completely presented?

Yes, but the underlying data have not been made available with the manuscript (as detailed below).

-Are the figures (Tables, Images) of sufficient quality for clarity?

Yes, however some modifications wold aid clarity (as detailed below).

Minor comments:

- Line 183: Please briefly define MIC50 & MIC90.

- Lines 208-210: Please consider providing a supplementary figure that stratifies the data by clinic to help illustrate the importance of this sampling. Similarly, please also consider stratifying the data by adult and child cohorts if the data are available to do so. 

- Figure 2: Please provide the denominator values for the samples examined each year (for the total of n=2725). This could potentially be done using a second y-axis and a line graph. Please also consider including the untested sequences as another colour in this plot to show the frequencies of missing AST data.

- Figure 3: Please define the blue lines & points in the figure legend. 

- Please make a spreadsheet of the data available as a supplementary file.

**Conclusions**

-Are the conclusions supported by the data presented?

-Are the limitations of analysis clearly described?

-Do the authors discuss how these data can be helpful to advance our understanding of the topic under study?

-Is public health relevance addressed?

Reviewer #1: The major conclusion is the fact that S. Paratyphi A isolates have a completely different pattern of AMR compared to S. Typhi, the former being mostly pan-susceptible to all commonly available antimicrobials and therefore these drugs could be used for empiric treatment of paratyphoid. But in addition, the high fluoroquinolone non-susceptible proportion of isolates clearly shows concern for across serovar transmission of resistance genes.

Reviewer #2: -Are the conclusions supported by the data presented?

Yes

-Are the limitations of analysis clearly described?

Yes

-Do the authors discuss how these data can be helpful to advance our understanding of the topic under study?

Yes 

-Is public health relevance addressed?

Yes

**Editorial and Data Presentation Modifications?**

Reviewer #1: N/A

Reviewer #2: - Line 57: ‘Year’ should be ‘years’.

- Line 85: It would give further context to the work if the authors could briefly state the current stage of vaccine development for S. Paratyphi A or refer to a recent review on the topic.

- Line 86/107: Please provide a citation for the statement regarding declining typhoid cases.

- Line 171: Please consider rephrasing this to refer to Dhaka rather than all of Bangladesh, unless suitable data from other regions support the notion of extrapolating these findings to Bangladesh more broadly.

- Line 224: ‘Majority’ should be ‘The majority’. Please also provide a citation for the lack of blood culture infrastructure in LMIC settings.

- Line 236: ‘acrb’ should be ‘acrB’.

- Line 235: ‘Resistance’ should be ‘resistant’.

- Line 240: It is unclear from the text at present how TCVs contribute to this. I think the intention is that when S. Typhi is adequately controlled with vaccines, empirical treatment for enteric fever could be based on Paratyphi A resistance patterns, but I am not sure. Please consider rephrasing this text to improve clarity.

**Summary and General Comments**

Reviewer #1: For the Paratyphoid cases from community and hospital setting,the improvement in rapid diagnostics to detect paratyphoid fever could allow for shifting of the empirical treatment of paratyphoid fever to the first line of drugs – ampicillin, cotrimoxazole, or chloramphenicol without affecting the efficacy of treatment. This would allow for preservation of reserve antibiotics including ceftriaxone and azithromycin tfor treatment of other severe infections.

Reviewer #2: Sajib and colleagues report on the trends in AMR from n=2,725 Salmonella Paratyphi A isolates collected over 23 years (1999-2021) from multiple clinics in Dhaka, Bangladesh. This manuscript fills a key knowledge gap regarding the burden of AMR Paratyphi A in Bangladesh, for which historical data are scant. A key finding is that despite popular older examples in the literature, Multidrug resistant S. Paratyphi A was not identified within the study period. The manuscript is well written, analyses appropriate, and the data are important for guiding empirical treatment of paratyphoid fever. The study is also timely given the rollout of typhoid conjugate vaccines in many endemic settings, which, as the authors note, may lead to an increase of paratyphoid cases. However, the manuscript could benefit from some minor edits to improve clarity, and the underlying data should be made available, as outlined in the other sections of this review.

PLOS authors have the option to publish the peer review history of their article (what does this mean?). If published, this will include your full peer review and any attached files.

Reviewer #1: Yes: Samuel Kariuki

Reviewer #2: No

Figure Files:

Data Requirements:

Reproducibility:

References

---

## [Decision Letter · Decision Letter 1]

15 Oct 2023

Dear Dr. Saha,

We are pleased to inform you that your manuscript 'Trends in antimicrobial resistance amongst Salmonella Paratyphi A isolates in Bangladesh: 1999-2021' has been provisionally accepted for publication in PLOS Neglected Tropical Diseases.

Best regards,

David Joseph Diemert, M.D.

Academic Editor

Stuart Blacksell

Section Editor

Reviewer's Responses to Questions

**Key Review Criteria Required for Acceptance?**

**Methods**

-Are the objectives of the study clearly articulated with a clear testable hypothesis stated?

-Is the study design appropriate to address the stated objectives?

-Is the population clearly described and appropriate for the hypothesis being tested?

-Is the sample size sufficient to ensure adequate power to address the hypothesis being tested?

-Were correct statistical analysis used to support conclusions?

-Are there concerns about ethical or regulatory requirements being met?

Reviewer #1: The objectives are clearly articulated and the revised version further clarifies issues raised during the initial review

Statistical analyses are also used to support the outcomes and conclusions from the study

Reviewer #2: Yes

**Results**

-Does the analysis presented match the analysis plan?

-Are the results clearly and completely presented?

-Are the figures (Tables, Images) of sufficient quality for clarity?

Reviewer #1: The study set out to investigate the prevalence and antimicrobial susceptibility of Paratyphi A in Bangladesh. The outcomes of this systematic surveillance over 23 years in multiple health care facilities highlights that barring fluoroquinolones, Paratyphi A has remained sensitive to most classes of antibiotics including the first line. Resistance to these drugs is usually carried on plasmids, and a recent large-scale genomic analysis of Salmonella Paratyphi A showed that the bacterium does not commonly carry such plasmids. The fact that these isolates remain nearly 95% susceptible to commonly available drugs makes a compelling case for empiric treatment of confirmed cases of Paratyphoid fever in Bangladesh.

Reviewer #2: Yes

**Conclusions**

-Are the conclusions supported by the data presented?

-Are the limitations of analysis clearly described?

-Do the authors discuss how these data can be helpful to advance our understanding of the topic under study?

-Is public health relevance addressed?

Reviewer #1: The conclusions are made with reference to the study outcomes. Authors have clearly indicated limitations that should be considered in drawing conclusions and parallels from these study outcomes

Reviewer #2: Yes

**Editorial and Data Presentation Modifications?**

Reviewer #1: (No Response)

Reviewer #2: Thank you for considering my comments, I am satisfied that my concerns have been addressed.

**Summary and General Comments**

Reviewer #1: This study data will be very helpful for epidemiologists and clinicians that see Paratyphi A patients in endemic settings. It is useful as a guide for selection of treatment options and also to enable wider surveillance in other settings as well

Reviewer #2: Thank you for considering my comments, I am satisfied that my concerns have been addressed.

PLOS authors have the option to publish the peer review history of their article (what does this mean?). If published, this will include your full peer review and any attached files.

Reviewer #1: **Yes: **Samuel Kariuki

Reviewer #2: No

---

## [Editor Report · Acceptance letter]

2 Nov 2023

Dear Dr. Saha,

We are delighted to inform you that your manuscript, "Trends in antimicrobial resistance amongst Salmonella Paratyphi A isolates in Bangladesh: 1999-2021," has been formally accepted for publication in PLOS Neglected Tropical Diseases.

Best regards,

Shaden Kamhawi

co-Editor-in-Chief

Paul Brindley

co-Editor-in-Chief
